# NLRP3-Dependent and -Independent Processing of Interleukin (IL)-1β in Active Ulcerative Colitis

**DOI:** 10.3390/ijms20010057

**Published:** 2018-12-23

**Authors:** Nicole Ranson, Mark Veldhuis, Brent Mitchell, Scott Fanning, Anthony L. Cook, Dale Kunde, Rajaraman Eri

**Affiliations:** 1School of Health Sciences, University of Tasmania, Launceston, Tasmania 7250, Australia; Dale.Kunde@utas.edu.au (D.K.); rderi@utas.edu.au (R.E.); 2Launceston General Hospital, Launceston, Tasmania 7250, Australia; Mark.Veldhuis@ths.tas.gov.au (M.V.); Brentlm@iinet.net.au (B.M.); Scott.Fanning@ths.tas.gov.au (S.F.); 3Wicking Dementia Research and Education Centre, Faculty of Health, University of Tasmania, Hobart, Tasmania 7000, Australia; Anthony.Cook@utas.edu.au

**Keywords:** NLRP3 inflammasome, Interleukin (IL)-1β, ulcerative colitis (UC), Crohn’s disease (CD), innate immune system

## Abstract

A contributing factor in the development of ulcerative colitis (UC) and Crohn’s disease (CD) is the disruption of innate and adaptive signaling pathways due to aberrant cytokine production. The cytokine, interleukin (IL)-1β, is highly inflammatory and its production is tightly regulated through transcriptional control and both inflammasome-dependent and inflammasome- independent proteolytic cleavage. In this study, qRT-PCR, immunohistochemistry, immunofluorescence confocal microscopy were used to (1) assess the mRNA expression of *NLRP3, IL-1β, CASP1* and *ASC* in paired biopsies from UC and CD patient, and (2) the colonic localization and spatial relationship of NLRP3 and IL-1β in active and quiescent disease. *NLRP3* and *IL-1β* were found to be upregulated in active UC and CD. During active disease, IL-1β was localized to the infiltrate of lamina propria immune cells, which contrasts with the near-exclusive epithelial cell layer expression during non-inflammatory conditions. In active disease, NLRP3 was consistently expressed within the neutrophils and other immune cells of the lamina propria and absent from the epithelial cell layer. The disparity in spatial localization of IL-1β and NLRP3, observed only in active UC, which is characterized by a neutrophil-dominated lamina propria cell population, implies inflammasome-independent processing of IL-1β. Consistent with other acute inflammatory conditions, these results suggest that blocking both caspase-1 and neutrophil-derived serine proteases may provide an additional therapeutic option for treating active UC.

## 1. Introduction

Ulcerative colitis (UC) and Crohn’s disease (CD) are characterized by chronic and relapsing inflammation of the gastrointestinal tract and are collectively known as inflammatory bowel diseases (IBD). Patients are usually first diagnosed during adolescence or early adulthood and in general present with symptoms of diarrhea, rectal bleeding, abdominal pain, and malnutrition [1]. IBD is regarded as a multifactorial disease process involving a combination of innate immune system defects and environmental triggers which occur in the genetically predisposed individual [2].

Important to innate immune defenses is the induction, production, and release of proinflammatory cytokines, which direct an effective host defense, and subsequent modulation of the adaptive immune response. One member of the IL-1 family of cytokines, IL-1β is considered highly inflammatory [3] and classified as a multifunctional cytokine. IL-1β participates in local and systemic responses to injury, infection, and inflammation. Clinically, IL-1β can evoke fever and hypotension [4], and control certain central nervous system functions such as sleep, pain, and appetite [5]. Locally, IL-1β can induce cytokine production, enhance T cell activation and antigen recognition, and direct neutrophils to the site of injury or infection [6,7,8]. 

IL-1β is produced as an inactive 31kDa proIL-1β protein in response to Toll-like receptor (TLR) activation and enzymatic cleavage is required to generate a bioactive 17kDa IL-1β fragment. The synthesis of a precursor cytokine requiring activation by proteases prevents aberrant secretion of IL-1β [9]. 

Several inflammasomes such as, NLRP1, NLRP3, NLRC4, NLRP6, NLRP7, NLRP12, AIM2 and IFI16 have been described for the canonical activation of caspase-1 and maturation of IL-1β [10]. Inflammasome activation is regarded as a two-step process. Transcription and translation of inflammasome components precedes the ligand activation step, which culminates in the assembly of the inflammasome platform and the maturation of IL-1β [11,12]. Important to innate immune defenses is the formation of the NLRP3 inflammasome which occurs in response to a wide range of microbial, environmental, and sterile ligands [10,13]. While the exact mechanisms of ligand activation are yet to be determined it is generally accepted that assembly of the NLRP3 inflammasome occurs in response to host derived factors altered by these agents. The indirect mechanisms include K^+^ efflux [14], phagolysosmal destabilization and release of cathepsins [15], the release of mitochondrial DNA or the mitochondrial phospholipid cardiolipin [16,17,18], translocation to the mitochondria [19,20,21] or the presence of mitochondrial derived reactive oxygen species (mROS) [19]. 

In addition to inflammasome-dependent production of IL-1β, several cell specific inflammasome-independent processes exist for the activation of IL-1β. IL-1β is a primary product of blood monocytes, tissue macrophages, neutrophils, dendritic cells and to a lesser extent, B lymphocytes and Natural Killer cells [22]. In blood monocytes, caspase-1 is constitutively expressed and the production of active IL-1β occurs via transcriptional control. This mechanism is thought to aid host defenses by reducing immune response time [23]. 

The zinc-dependent metalloproteinases, meprin A and meprin α are highly expressed at the brush-border membranes of the kidney and intestine, and can generate biologically active IL-1β [24]. During acute inflammation, neutrophil and macrophage derived serine proteases, proteinase 3, elastase and cathepsin G can process proIL-1β to biologically active IL-1β [23,25,26,27]. Indeed, microbes such as *Candida albicans* and *Staphylococcus spp.* can themselves produce proteases, which possess the ability to cleave proIL-1β [28,29]. Despite this, the dependence of NLRP3 to IL-1β production has not been investigated in human UC or CD biopsy material, and therefore the aim of this study was to investigate the NLRP3 inflammasome and its cleavage product, IL-1β in active disease.

## 2. Results

### 2.1. Clinical Characteristics of IBD Cohort

The control group for this study included mucosal biopsies from 20 healthy patients who underwent routine colonoscopy investigations. The mean age of the control group was 57 ± 14 and comprised 11 females and 9 males. 

The UC cohort included 44 patients, of these 27 (61%) were taking prescription medication to control their disease while 17 patients (39%) were untreated. Most UC patients were prescribed 5-ASAs 81% (22/27) with 50% (11/22) of 5-ASAs users also taking a defined tapering course of corticosteroids for an active flare of disease (Table 1). 

The CD cohort included 21 patients, of these 16 patients (76%) were being treated with prescription medication, while 5 patients (24%) were untreated at time of biopsy collection. Medication regimes favored 5-ASAs (29%, 6/21) and immunomodulators (29%, 6/21) while there was less use of corticosteroids (10%, 2/21) and biologic agents (19%, 4/21) (Table 2). As expected, there was a higher proportion of smokers in the CD group (24%, 5/21) when compared to the UC group (5%, 2/44).

### 2.2. NLRP3 and IL-1β Are Upregulated in Active Disease

Using paired human colonic biopsies from quiescent and active UC and CD, we began by investigating the mRNA expression of *NLRP3* inflammasome related genes using quantitative real-time-polymerase chain reaction (qRT-PCR). The expression of *NLRP3* and *IL-1β* were found to increase with disease activity in both UC and CD (Table 3, Figure 1A,B). Consistent with NLRP3 inflammasome activation the expression of *CASP1* and *ASC* also increased (Table 3, Figure 1C,D). Correlation analysis of mRNA expression revealed a stronger correlation with IL-1β in active CD (Rs = 0.96, *p* < 0.01) (Figure 1F) than in active UC (Rs = 0.78, *p* < 0.01) (Figure 1E). 

### 2.3. NLRP3 is Localized to the Influx of Lamina Propria Cells in Active UC

We next sought to determine the cellular localization of NLRP3 and IL-1β in active and remission IBD by using immunohistochemistry. The expression of NLRP3 was found to increase with disease activity in both active UC and active CD. For active UC, prominent cytoplasmic staining was evident in the neutrophils and other immune cells of the lamina propria while diffuse staining was evident in the epithelial cell region. Consistent with pathology reports, biopsy sections from active UC patients were characterized by an influx of lamina propria immune cells concentrated around regions of crypt distortion or mucosal inflammation (Figure 2, black dotted box indicates increased numbers of lamina propria neutrophils). 

For active CD, the reported influx of lamina propria immune cells seen in active UC was not evident; however, the intensity of staining for NLRP3 within the lamina propria immune cells was comparable. Moderate cytoplasmic staining for NLRP3 was also evident in the epithelial cell layer including around goblet cells.

In the normal colon, remission UC and remission CD only scattered NLRP3 staining was evident in a lamina propria immune cells. Quantitative analysis of immunohistochemistry staining confirmed the increased expression of NLRP3 in active UC (*p* < 0.001) and active CD (*p* = 0.007) (Figure 4A). Interestingly, expression of NLRP3 in remission UC and remission CD are similar to that observed in the normal colon.

In normal colon biopsies, high expression of IL-1β was observed within the epithelial cell layer of the mucosal crypts, including the cytoplasm of intestinal goblet cells. For active UC and active CD high IL-1β expression was predominantly localized to the immune cells of the lamina propria and scattered within the epithelial cell layer (Figure 3). Quantitative analysis confirmed the increased expression of IL-1β in active UC (*p* < 0.001) and active CD (*p* = 0.006). Notably, the expression of IL-1β in remission UC was less than that observed in the normal colon (*p* = 0.003) (Figure 4B).

### 2.4. The Contribution of NLRP3 to IL-1β Is Reduced in Active UC

The concurrent expression of NLRP3 and IL-1β within the lamina propria cell populations in active disease prompted a detailed colocalization analysis using immunofluorescence confocal microscopy. Manders coefficients (M1 and M2) were used to evaluate the reciprocal association ratio between fluorescence markers with values ranging from 0.5 to 1.0 indicating a positive association. Pearson’s correlation coefficient was used to describe the correlation of the intensity distribution between channels. Values range from -1 to 1 and indicate the strength of the negative or positive correlation [30]. For normal colon, a high proportion of colocalization between NLRP3 and IL-1β was evident within the lamina propria immune cells (Figure 5).

Quantification of Manders coefficients confirmed a positive association between NLRP3 and IL-1β (M1, NLRP3:IL-1β = 0.81 ± 0.06; M2, IL-1β:NLRP3 = 0.82 ± 0.06) indicating a similar contribution of one to the other (Figure 6A).

Manders coefficients demonstrated less colocalization of NLRP3 to IL-1β in active UC (M1, NLRP3:IL-1β = 0.67 ± 0.04; M2, IL-1β:NLRP3 = 0.61 ± 0.09) (Figure 6B) when compared to normal colon biopsies. Interestingly, the appearance of a y-dominated scattergram suggests increased IL-1β production without concomitant and spatially coincident increases in NLRP3 (Figure 5). The lack of correlation between NLRP3 and IL-1β in active UC is confirmed by the near zero (0.03) Pearson correlation coefficient (Figure 6D).

For active CD, the colocalization of NLRP3 and IL-1β (M1, NLRP3: IL-1β = 0.78 ± 0.05; M2, IL-1β:NLRP3 = 0.80 ± 0.05; Pearson’s correlation coefficient = 0.29) was similar to that seen in the normal colon (Figure 6C,D).

## 3. Discussion

This study demonstrates the upregulation of *NLRP3* and *IL-1β* in active IBD and describes the colonic localization of NLRP3 and IL-1β in active and remission disease. Furthermore, the increase in IL-1β production without concomitant and spatially coincident increases in NLRP3 observed only in active UC, and in a background of a dominated neutrophilic lamina propria cell population suggests inflammasome-independent processing of IL-1β. 

The inflammasome complex is a key regulator of IL-1β production. The demonstrated increase in *IL-1β* gene expression in both active UC and CD is well established and consistent with previous research in human derived material [31,32,33,34,35]. In normal colon, IL-1β was localized to the epithelial layer and generally absent from the lamina propria immune cells, while NLRP3 expression was generally present at low levels in the lamina propria immune cells. With disease activity, the expression of IL-1β shifted from the epithelial cell layer to the lamina propria cells and the expression of both NLRP3 and IL-1β in the lamina propria immune cells intensified. Previously, NLRP3 expression has been demonstrated in granulocytes, such as neutrophils, dendritic cell, monocytes, epithelial cells, T cells and B cells, with subcellular distribution localization mainly in the cytoplasm [36]. 

We observed increased numbers of neutrophils in the lamina propria of active UC patients which was consistent with pathology histology reports. The composition of the lamina propria infiltrates, microscopic architectural abnormalities and inflammatory changes are all important features for discriminating normal mucosa from IBD [37]. During active UC, neutrophils dominate the lamina propria immune cell population and are often the effector cells surrounding epithelial damage or mucosal inflammation [38]. Consequently, many grading scales for histological assessment of inflammation in UC include the presence and locality of neutrophils [39,40,41]. Furthermore, the use of neutrophils as a marker of disease activity in UC is supported by leucocyte scanning studies [42,43]. 

The lack of acceptable sensitivity, specificity, and reproducibility discourages diagnosis of CD based on the lamina propria infiltrate. Diagnosis of CD therefore relies on the presence of epithelioid granuloma, relatively unchanged crypts or segmented distribution of crypt atrophy, crypt distortion together with discontinuous focal or patchy inflammation (skip lesions) and mucin preservation in the epithelium at an ulcer edge. Focal inflammation is often characterized by a small collection of inflammatory cells in otherwise normal mucosa [44,45,46]. Microscopic structural abnormalities and the influx of lamina propria immune cells are not features of normal mucosa.

We report reduced colocalization between NLRP3 and IL-1β in active UC when compared to active CD and normal mucosa biopsies, suggesting NLRP3 inflammasome-independent mechanisms contribute more to the overall IL-1β production in active UC. Inflammasome-independent processing of proIL-1β in acute inflammatory conditions is well established and has been shown to occur where neutrophils dominate the lamina propria cell populations. For instance, host resistance to disseminated candidiasis is provided by neutrophil-derived proteinase 3 activation of IL-1β and not caspase-1 activation [29,47,48]. Similarly, during acute arthritis, neutrophil-derived proteinase 3 plays a dominant role in in the production of bioactive IL-1β [49]. Again, serine proteases from neutrophils drive the inflammation and production of mature IL-1β in a murine model of osteomyelitis [50,51].

The blockade of NLRP3 using selective small-molecule inhibitors has recently been the focus of much research. MCC950 has been shown to inhibit canonical and non-canonical NLRP3 inflammasome activation and reduce IL-1β production [52]. However, based on the results from this study, in diseases such as UC where neutrophils dominate the lamina propria infiltrate, blocking NLRP3 or caspase-1 may not reduce in vivo IL-1β levels. 

In summary, the results of this study suggest that during normal gut homeostasis inflammasome-dependent caspase-1 is a major contributor to IL-1β production. However, during active UC when neutrophils dominate the lamina propria maturation of IL-1β by neutrophil deriver serine proteases may contribute more to IL-1β production than inflammasome-dependent caspase-1 cleavage.

The dual blockage of both caspase-1 and proteinase 3 is seen as a potential anti-inflammatory therapeutic option in a murine model of arthritis [49]. Likewise, future IBD research should now focus on assessing neutrophil-derived IL-1β production and the therapeutic potential of blocking both caspase-1 and neutrophil-derived serine proteases in active UC. 

## 4. Materials and Methods

### 4.1. Ethics

The study was conducted at the University of Tasmania, in collaboration with local gastroenterologists, St Vincent’s Private Hospital and Launceston General Hospital. Human studies were approved by the Human Research Ethics Committee, Tasmania (approval number: H11930, 19 October 2011). All participant in this study provided informed written consent or parental consent and were aged between 15 and 80 years. All authors had access to the study data, reviewed, and approved the final manuscript.

### 4.2. Study Participants and Biopsy Collection

A total of 85 patients presenting for routine colonoscopy investigations between June 2012 and June 2014 were recruited for this study. IBD patients were excluded if they were experiencing co-existing irritable bowel syndrome (IBS) or non-IBD associated gastrointestinal bleeding at the time of colonoscopy. IBD patients had paired colonic biopsies taken from both inflamed mucosa and non-inflamed mucosa. Control patients were excluded if they had a previous history of IBD or IBS. 

Of the 85 participants, 30 UC patients and 15 CD patients had colonic biopsies taken from both inflamed mucosa and non-inflamed mucosa. Ten UC patients and four CD patients were in remission at the time of colonoscopy and only had biopsies taken from non-inflamed mucosa. Four UC and two CD patients presented with pancolonic disease and had biopsies taken from inflamed mucosa only. Twenty control patients had biopsies collected from healthy non-inflamed mucosa. 

Disease assessment and biopsy location was at the discretion of the treating gastroenterologist (Mark Veldhuis, Brent Mitchell and Scott Fanning). Selection for immunohistochemistry or immunofluorescence microscopy analysis was based on patient pathology reports and only biopsies from the descending left colon with active disease features were chosen. 

### 4.3. Gene Expression Analysis

Total RNA was extracted from tissue biopsies using the RNeasy Plus Mini Kit (QIAGEN, Venlo, Netherlands). RNA quality and concentration were determined using the Experion automated electrophoresis system (BIO-RAD, Hercules, CA, USA). RNA was reverse transcribed to cDNA using the iScript cDNA synthesis kit (BIO-RAD, Hercules, CA, USA). Quantitative real-time-polymerase chain reaction (qRT-PCR) was performed on a StepOne analyzer (Applied Biosystems, CA, USA) using a TaqMan Universal PCR Master mix (Applied Biosystems, CA, USA) and on-demand gene-specific primers for human *IL-1β*, *NLRP3*, *CASP1* and *ASC*. Gene expression was quantified using the comparative (ΔΔCt) method where the threshold cycle (C_T_) was normalized to the reference gene, eukaryotic translation elongation factor (*EEF2*).

### 4.4. Immunohistochemistry Analysis

Immunohistochemistry was performed on 10% (*v/v*) formalin fixed, paraffin embedded left colon biopsies. Sections were deparaffinized and subjected to 0.01 M citrate buffer (pH 6.0) antigen retrieval at 121 °C for 4 min in a decloaking chamber. Endogenous peroxidases were quenched by a 5-min incubation in 10% H_2_O_2_ in methanol. Non-specific binding was blocked by a 20-min incubation in BioCare background Sniper solution (BS966G, BioCare, Concord, CA21520, USA). Immunostaining was performed using specific antibodies against IL-1β (ab9722, Abcam, Cambridge, MA, USA) and NLRP3 (ab17267, Abcam, Cambridge, MA, USA) at room temperature for 1 h. An additional incubation with MACH 1 mouse probe (UP537L10, BioCare, Concord, CA21520, USA) for 15 min at room temperature was performed on sections incubated with the anti-NLRP3 antibody. Samples were then incubated with a horseradish peroxidase (HRP)-polymer (MRH53BL10, BioCare) for 30 min, stained with Betazoid DAB chromogen (BDB900B, BioCare, Concord, CA21520, USA), counterstained with hematoxylin, dehydrated and mounted with distyrene, plasticizer and xylene. Slides were examined using an IX71 microscope (Olympus Australia, Melbourne, Australia). 

### 4.5. Immunofluorescence confocal Microscopy

Immunofluorescence staining was performed on 10% *v/v* formalin fixed, paraffin embedded left colon biopsies. Sections were deparaffinized and subjected to 0.01 M citrate buffer (pH 6.0) antigen retrieval at 121 °C for 4 min in a decloaking chamber. Non-specific binding was blocked by a 1-h incubation in blocking buffer [0.1 mol/L phosphate-buffered saline (PBS)/5% normal goat serum/0.05% Tween-20] at room temperature in the dark. Immunofluorescence staining was performed using specific antibodies against NLRP3 (ab16097, Abcam, Cambridge, MA, USA) and IL-1β (ab9722, Abcam, Cambridge, MA, USA) in the dark, at room temperature for 1 h or alternatively overnight at 4 °C. Sections were then incubated for 1 h in the dark with one or more Alexa Fluor conjugated-secondary antibodies (Cell Signaling Technology, Danvers, MA, USA). Section were incubated with 4′,6-diamidino-2-phenylindole Dihydrochloride (DAPI, ThermoFisher Scientific, Waltham, MA, USA) diluted in PBS and mounted with ProLong Gold Antifade (P36930, ThermoFisher Scientific, Waltham, MA, USA). Slides were examined using a FV1200 Laser Scanning Confocal Inverted Microscope (Olympus Australia, Melbourne, Australia). 

### 4.6. Data Analysis

Analysis of qRT-PCR data from colon biopsies was performed using Microsoft Excel and GraphPad Prism software (version 7.0, GraphPad Software Inc. CA, USA). Colon biopsy gene expression was compared to a healthy control group and normalized to the housekeeping gene EEF2. Group difference were tested using paired t tests of log (10) transformed data. Data are presented as Median Relative Expression (R.E) and interquartile ranges. 

Quantitative analysis of immunohistochemistry data was performed using the FIJI version (Dec 2009) of ImageJ software (http://imagej.nih.gov/ij/downloads.html, Rasband, W.S., ImageJ, U.S. National Institutes of Health, Bethesda, MD, USA), Microsoft Excel and GraphPad Prism (version 7, GraphPad Software Inc. CA, USA). Briefly, the mean intensity of the DAB signal was measured in FIJI, optical density was calculated in Excel using log (maximum intensity/mean intensity), where the maximum intensity of an 8-bit image = 255. Images containing partial tissue were excluded from analysis because of their potential to bias results by reducing the average optical density. Group differences and statistical significance were evaluated using a one-way analysis of variance (ANOVA) followed by Dunnett’s multiple comparison. Data are presented as mean ± standard deviation. 

Colocalization analysis was performed using the Coloc 2 plugin in the FIJI version (Dec 2009) of ImageJ software (http://imagej.nih.gov/ij/downloads.html, Rasband, W.S., ImageJ, U. S. National Institutes of Health, Bethesda, Maryland, USA). To ensure consistency, measurements were only performed on biological relevant regions of interest (ROIs) contained within 400X double stained, background corrected, confocal images. Manders correlation coefficients (M1 and M2) were used to determine the co-occurrence of the fluorescence channels and the Pearson’s correlation coefficient (above the threshold) was used to describe the correlation of the intensity distribution between channels. 2D Histograms generated in Coloc 2 provide an overview of the fluorescent relationship between channel intensities for homologous pixels. Colocalization data are presented as mean ± standard deviation. Statistical significance of Pearson’s correlation for the NLRP3/IL-1β double staining was evaluated in GraphPad Prism (version 7, GraphPad Software Inc. CA, USA), using one-way ANOVA, followed by Tukey’s multiple comparisons test. In all presented data the significance threshold was set at *p* < 0.05.

## Figures and Tables

**Figure 1 ijms-20-00057-f001:**
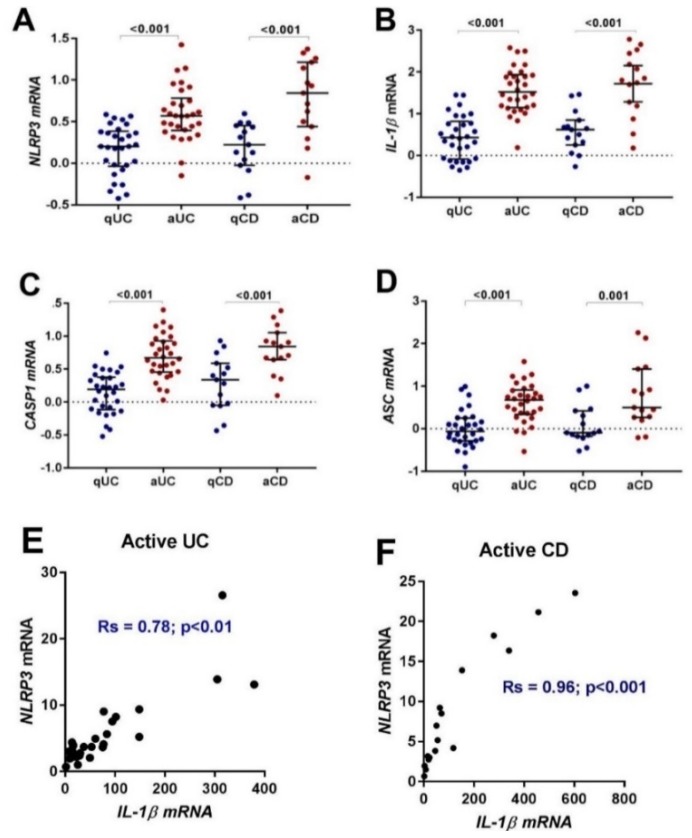
Gene expression and correlation of NLRP3 inflammasome related genes in paired biopsies from quiescent UC (qUC), active UC (aUC), quiescent CD (qCD) and active. CD (aCD). The mRNA expression of (**A**) *NLRP3*, (**B**) *IL-1β*, (**C**) *CASP1*, and (**D**) *ASC* was compared to a healthy control group and normalized to the housekeeping gene, *EEF2.* Individual patient results are shown as dots. Horizontal lines indicate the median relative expression (R.E) and error bars represent the interquartile ranges. Group differences were tested using Mann-Whitney test of log_10_ transformed data. E) The correlation of NLRP3 and IL-1β mRNA in active UC. **F)** The correlation of NLRP3 and IL-1β mRNA in active UC. The significance threshold was *p* < 0.05.

**Figure 2 ijms-20-00057-f002:**
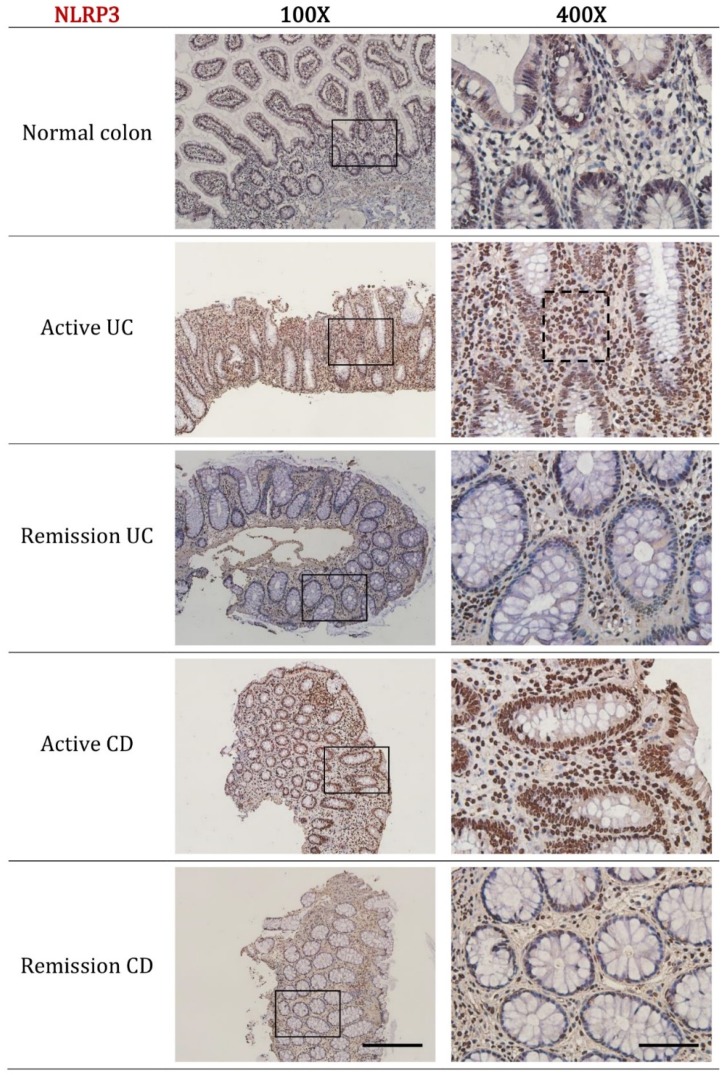
Representative immunohistochemistry images of NLRP3 expression in normal colon, active UC, remission UC, active CD and remission CD. All biopsies were taken from the left colon (unless otherwise stated as ileum biopsy), paraffin embedded, cut into 5 μm sections and incubated with NLRP3 (ab17267, Abcam, Cambridge, MA, USA) at a dilution of 1:300. Disease diagnosis was confirmed by a practicing pathologist. Black dotted box indicates increased numbers of lamina propria neutrophils in active UC. Scale bars represent 200 μm for 100× and 50 μm for 400× magnification.

**Figure 3 ijms-20-00057-f003:**
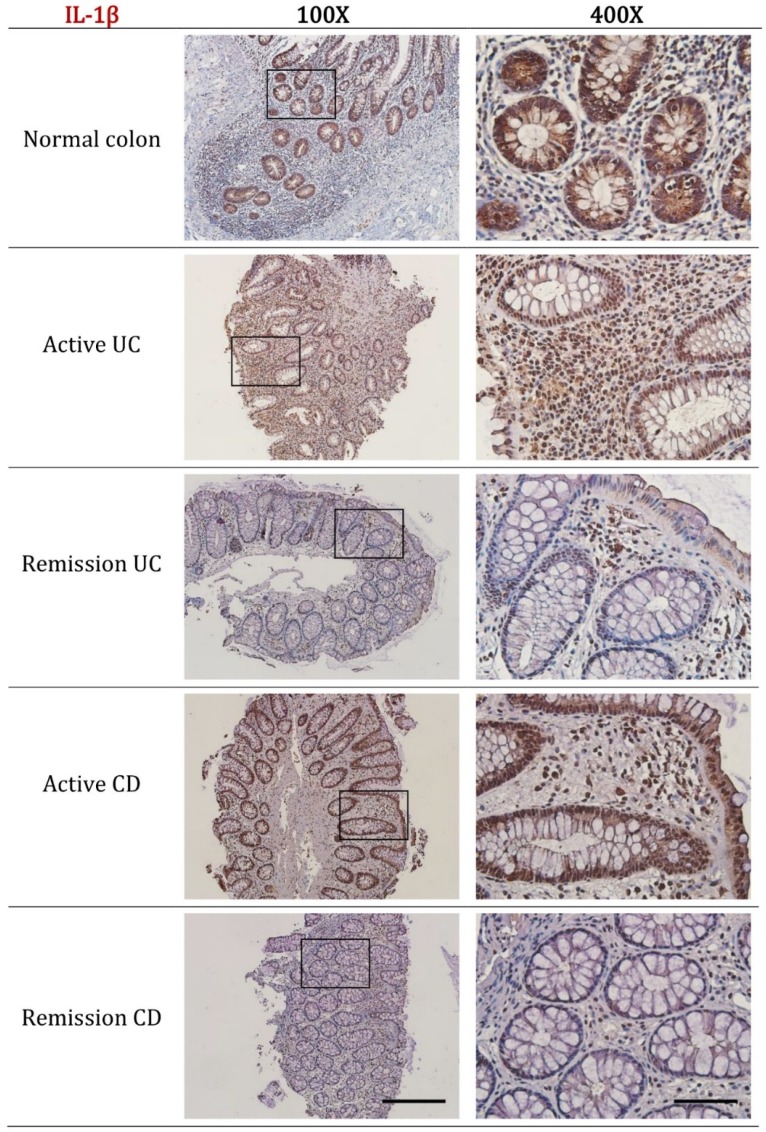
Representative immunohistochemistry images of IL-1β expression in normal colon, active UC, remission UC, active CD and remission CD. All biopsies were taken from the left colon (unless stated as ileum biopsy), paraffin embedded, cut into 5 μm sections and incubated with IL-1β (ab9722, Abcam, Cambridge, MA, USA) at a dilution of 1:300. Disease diagnosis was confirmed by a practicing pathologist. Scale bars represent 50 μm for 400× and 200 μm for 100× magnification.

**Figure 4 ijms-20-00057-f004:**
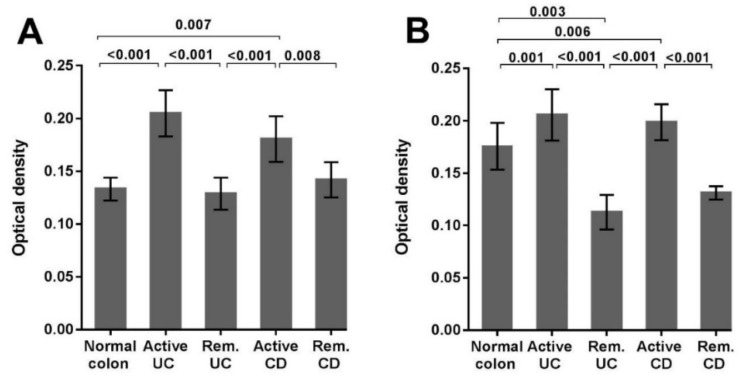
Quantification of NLRP3 and IL-1β IHC staining in biopsies sections from normal and IBD patients. (**A**) Quantitative analysis of NLRP3 expression in sections from normal colon (total number of images analyzed = 10), active UC (*n* = 91), remission UC (Rem. UC, *n* = 21), active CD (*n* = 42), and remission CD (Rem. CD, *n* = 18). (**B**) Quantitative analysis of IL-1β expression in sections from normal colon (*n* = 24), active UC (*n* = 46), remission UC (Rem. UC, *n* = 20), active CD (*n* = 64), and remission CD (Rem. CD, *n* = 18). Paraffin embedded left colon mucosal biopsies were analyzed by immunohistochemistry and the optical intensity of DAB staining due to primary antibody was determined using FIJI software. All data are presented as mean ± standard deviation. Statistical significance was evaluated using Dunn’s multiple comparison one-way analysis of variance (ANOVA). The significance threshold was *p* < 0.05.

**Figure 5 ijms-20-00057-f005:**
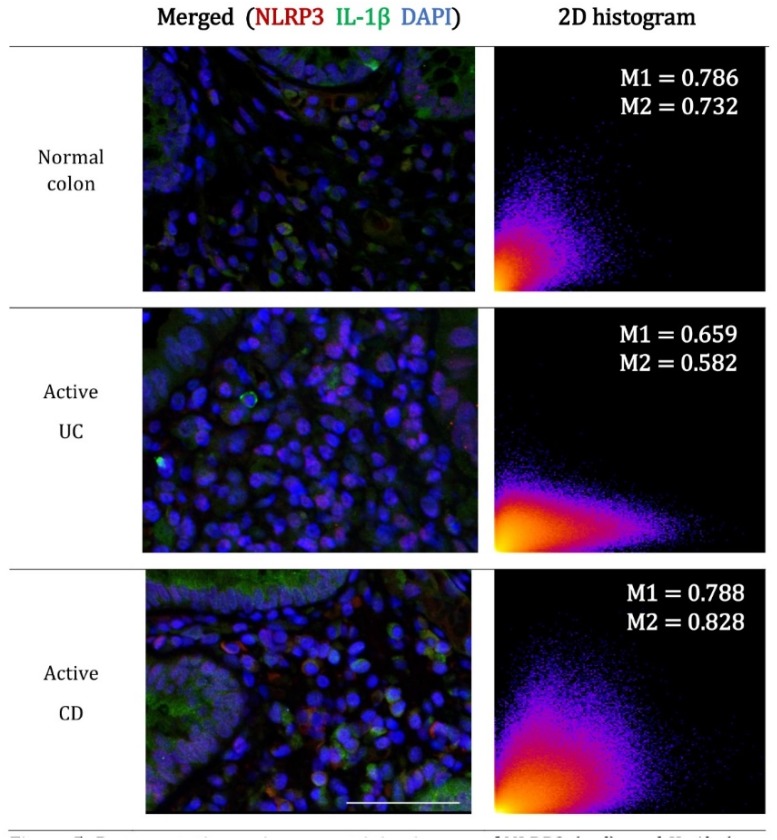
Representative co-immunostaining images of NLRP3 (red) and IL-1b (green) in normal colon and active UC and CD. Parrafin embedded colon biopsies were simultaneously stained with NLRP3 (ab16097, Abcam, Cambridge, MA, USA) and IL-1β (ab9722, Abcam, Cambridge, MA, USA) and visualized using AlexaFluor 647 conjugated mouse anti-goat (red) and AlexaFluor 555 conjugated rabbit anti-goat (green) respectively. Nuclei were stained with 4′,6-diamidino-2-phenylindole (DAPI, blue). The 2D histogram visualizes the overall relationship of channel intensities for homologous pixels. The coordinates of the scattergram are the channel (CH) intensities of NLRP3 (red-CH1-x) and IL-1β (green-CH2-y) within each pixel. The value at each location indicates the incidence of the combination. Scale bar = 50 μm for 400× magnification.

**Figure 6 ijms-20-00057-f006:**
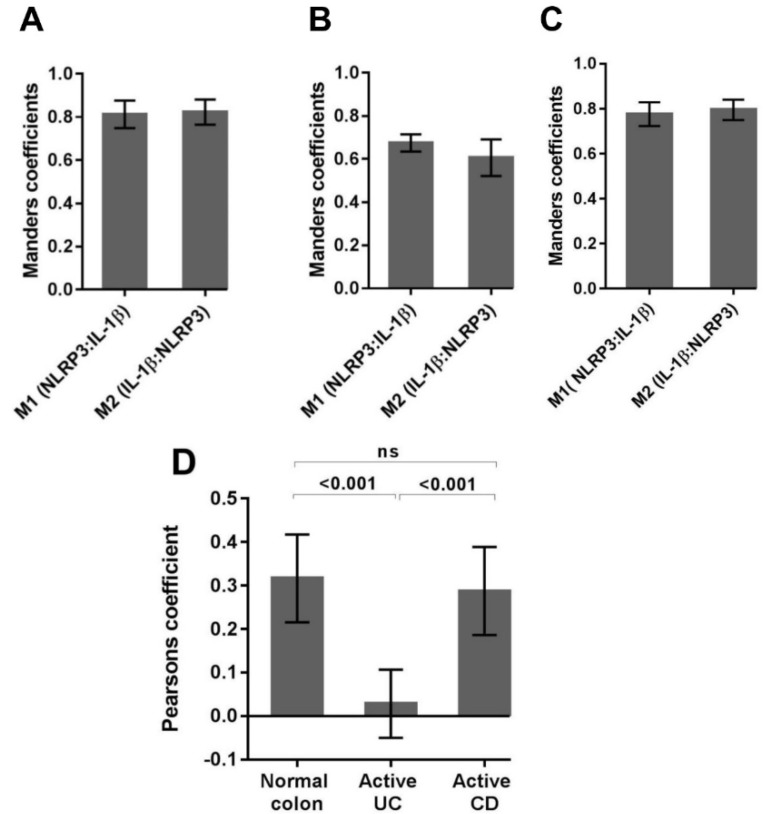
The colocalization of NLRP3 and IL-1β in normal and active disease as indicated by Manders colocalization coefficients and Pearson’s correlation coefficients. Each data set was compiled from lamina propria regions of interest (ROI), 400× images from (**A**) normal colon (*n* = 20), (**B**) active UC (*n* = 20), (**C**) active CD (*n* = 20). Manders coefficients (M1 and M2) were used to evaluate the reciprocal association ratio between fluorescence markers. (**D**) The Pearson’s correlation coefficient describes the fluorescence intensity distribution between the channels. Averages are presented as mean + standard deviation. Significance was tested using one-way ANOVA followed by Tukey’s multiple comparison test. The significance threshold was *p* < 0.05.

**Table 1 ijms-20-00057-t001:** Clinical characteristics of ulcerative colitis patients.

Variable	Patients with Active Disease		Patients in Remission
	Biopsies Taken from both Active and Inactive Regions	Biopsies Taken from Active Regions only	Biopsies Taken from Inactive Regions
Patients, *n*	30	4	10
Age (years)	50 ± 18	39 ± 21	49 ± 11
Gender			
female	16	2	6
male	14	2	4
Smoker, *n* (%)	1 (3)	0	1 (10)
Disease extent *			
Proctitis only	7 (23)	-	-
Proctosigmoiditis	14 (47)	-	-
Left-sided colitis	8 (27)	-	-
Pancolitis	1 (3)	4 (100)	4 (100)
Current Medication for UC			
Untreated	12 (40)	2 (50)	3 (30)
Aminosalicylates (5-ASA)	16 (53)	1 (25)	5 (50)
Corticosteroids	6 (20)	2 (50)	3 (30)
Immunomodulator therapy	4 (13)	-	-
Topical therapy	0	-	-
Biologic therapy	0	-	-
Antibiotics/probiotics/vitamin supplements	6	-	1

Age and disease duration data are shown as mean value ± SD. Other data indicates the number of patients. Group percentages are shown in brackets. * Disease Extent: Proctitis: Disease only in the rectum; Proctosigmoiditis: Disease in the rectum and sigmoid colon; Left-sided colitis: Limited or distal colitis. Disease in left side of the colon; Pancolitis: Disease in the entire colon.

**Table 2 ijms-20-00057-t002:** Clinical characteristics of Crohn’s disease patients.

Variable	Patients with Active Disease		Patients in Remission
	Biopsies Taken from both Active and Inactive Regions	Biopsies Taken from Active Regions only	Biopsies Taken from Inactive Regions
Patients, *n*	15	2	4
Age (years)	44 ± 16	37 ± 13	20 ± 7
Gender			
female	7	1	3
male	8	1	1
Smoker, *n* (%)	4 (27)	0	1 (25)
Disease extent *			
Ileum only (L1)	4 (27)	1	-
Ileum and colon (L3)	2 (13)	-	-
Colon only (L2)	9 (60)	1	-
Extra-intestinal manifestations (e.g., ileocecal/perianal/proctitis)	4 (27)	-	-
Previous surgical resection	4 (27)	-	1
Current Medication for CD			
Untreated	4 (27)	-	1 (25)
Aminosalicylates (5-ASA)	5 (33)	-	1
Corticosteroids	2 (13)	-	-
Immunomodulator therapy	6 (40)	1	1
Topical therapy	-	-	-
Biologic therapy	2 (13)	1	1
Antibiotics/probiotics/vitamin supplements	-	-	-

Age and disease duration data are shown as mean value ± SD. Other data indicates the number of patients. Group percentages are shown in brackets. Montreal CD classifications; L1, terminal ileum involvement; L2, colonic disease; L3 ileocolonic involvement. * Disease Extent: Extra-intestinal manifestations (e.g., ileocecal/perianal/proctitis) included as a separate section but can co-exist with colonic and ileum manifestations.

**Table 3 ijms-20-00057-t003:** Relative expression of NLRP3 related genes in IBD.

TargetGene	Disease Phenotype	Quiescent Disease		Active Disease		*p*
		Median R.E	IQR	Median R.E	IQR	
*NLRP3*	UC ^#^	1.5	1.0–2.3	3.7	2.5–5.7	<0.001
	CD ^	1.6	0.9–2.8	5.2	2.9–15.1	<0.001
*IL-1β*	UC ^#^	2.2	0.8–5.9	28.1	13.2–78.7	<0.001
	CD ^	4.1	1.5–7.0	55.1	17.2–215.7	<0.001
*CASP1*	UC ^#^	1.3	0.8–2.3	5.1	2.9–9.3	<0.001
	CD ^	2.5	0.9–3.9	6.9	4.5–10.2	<0.001
*ASC*	UC ^#^	1.0	0.5–1.7	4.8	2.3–9.2	<0.001
	CD ^	1.3	0.7–2.8	3.5	1.9–16.9	0.001

Quantitative RT-PCR was used to determine the expression of inflammasome related genes in IBD. Median Relative Expression (R.E) and interquartile ranges (IQR) are relative to the control group and normalized to the housekeeping gene EEF2. Group differences were tested using Mann-Whitney test of log (10) transformed data. The significance threshold was *p* < 0.05. ^#^ Column statistics for UC patients; quiescent disease, *n* = 40, active disease, *n* = 34; group difference, *n* = 30, ^ Column statistics for CD patients; quiescent disease, *n* = 19; active disease, *n* = 17, Group differences, *n* = 15.

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
