# Peer review of "NLRP3-Dependent and -Independent Processing of Interleukin (IL)-1β in Active Ulcerative Colitis"

_ijms, 2018, doi:10.3390/ijms20010057_

Round 1

Reviewer 1 Report

The manuscript by Ranson et al., investigates the expression levels of NLRP3, IL-1β, CASP1 and ASC in biopsies from UC and CD patient and the localisation of NLRP3 and IL-1β in active and quiescent disease. The authors showed that NLRP3 and IL-1β are upregulated in both UC and CD. In active disease, NLRP3 was consistently expressed within the neutrophils and other immune cells of the lamina propria, whereas IL-1β was localised to the infiltrate of lamina propria immune cells. The disparity in spatial localisation of IL-1β and NLRP3 was observed only in active UC. These results suggest an inflammasome-independent processing of IL-1β in active UC.

It is a well-written manuscript. The results are presented and discussed efficiently, and the methodologies are described adequately. Importantly, the study was based in a large number of UC and CD patients, and detailed medical records were provided.

Although the study is descriptive and lacks in depth analysis it is an interesting work and the manuscript is suitable for publication.

The authors should discuss more the observed differences in spatial localisation of IL-1β and NLRP3 between active UC and active CD.

Author Response

Response to Reviewer 1 Comments

Point :The manuscript by Ranson et al., investigates the expression levels of NLRP3, IL-1β, CASP1 and ASC in biopsies from UC and CD patient and the localisation of NLRP3 and IL-1β in active and quiescent disease. The authors showed that NLRP3 and IL-1β are upregulated in both UC and CD. In active disease, NLRP3 was consistently expressed within the neutrophils and other immune cells of the lamina propria, whereas IL-1β was localised to the infiltrate of lamina propria immune cells. The disparity in spatial localisation of IL-1β and NLRP3 was observed only in active UC. These results suggest an inflammasome-independent processing of IL-1β in active UC.

It is a well-written manuscript. The results are presented and discussed efficiently, and the methodologies are described adequately. Importantly, the study was based in a large number of UC and CD patients, and detailed medical records were provided.

Although the study is descriptive and lacks in depth analysis it is an interesting work and the manuscript is suitable for publication.

The authors should discuss more the observed differences in spatial localisation of IL-1β and NLRP3 between active UC and active CD.

Response: The reviewer comments are appreciated and based on these comments we have expanded the text from lines 132 – 164 to describe in more detail the differences in spatial localisation of IL-1β and NLRP3 in active disease. The text now reads….

Line 132:… 2.3 NLRP3 is localized to the influx of lamina propria cells in active UC

We next sought to determine the cellular localisation of NLRP3 and IL-1β in active and remission IBD by using immunohistochemistry. The expression of NLRP3 was found to increase with disease activity in both active UC and active CD. For active UC, prominent cytoplasmic staining was evident in the neutrophils and other immune cells of the lamina propria while diffuse staining was evident in the epithelial cell region. Consistent with pathology reports, biopsy sections from active UC patients were characterised by an influx of lamina propria immune cells concentrated around regions of crypt distortion or mucosal inflammation (Figure 2, black arrow indicates increased numbers of lamina propria neutrophils).

For active CD, the reported influx of lamina propria immune cells seen in active UC was not evident, however the intensity of staining for NLRP3 within the lamina propria immune cells was comparable. Moderate cytoplasmic staining for NLRP3 was also evident in the epithelial cell layer including around goblet cells.

In the normal colon, remission UC and remission CD only scattered NLRP3 staining was evident in a lamina propria immune cells. Quantitative analysis of IHC staining confirmed the increased expression of NLRP3 in active UC (P<0.001) and active CD (P=0.007) (Figure 4A). Interestingly, expression of NLRP3 in remission UC and remission CD are similar to that observed in the normal colon.

In normal colon biopsies, high expression of IL-1β was observed within the epithelial cell layer of the mucosal crypts, including the cytoplasm of intestinal goblet cells. For active UC and active CD high IL-1β expression was predominantly localized to the immune cells of the lamina propria and scattered within the epithelial cell layer (Figure 3).

Reviewer 2 Report

The authors analysed the expression of IL-1beta and NLRP3 in biopsies from IBD patients by qPCR analysis. They found upregulation of both in active UC and CD. Furthermore, immunostaining of colon paraffin sections were stained for NLRP3 and IL-1beta. Here, the quantification revealed a higher number of double+ cells in active UC as well as active CD. Less colocalisation is observed with NLRP3 and IL-1beta in active UC in comparison to normal and active CD patient sections.

First, the manuscript is more descriptive then explanatory. The authors should more try to undercover the mentioned blockade of caspase1 or proteinase-3 in relation with their patient biopsies.

The house keeping gene EEF2 is used in Figure1. From literature it is known that this gene is stable in experimental models if colitis, but in human colon material it is rarely used. Why did this gene was used ?

The quantification in Figure 2 is difficult to verify. Did the authors mean the blue shadowed cells or the brown cells ? The arrow shows somewhere in the section. What about the positive cells in the other sections ? Some more arrows should help to identify the positive cells. Same is concerning the Figure 3. Here no arrows were visible. Are some positive cells can be identified ?

Helpful would be, if the authors can add the negative control stainings to the figures.

In Figure 4, the optical density should be further explained. What unit do OD have ? How is the OD been calculated ?

Concerning the Figure 5, the staining of NLRP3 is not optimal done. I would expect the signal not extracellular than intracellular. Here may be a positive control as well as a negative control would help to identify the right cells.

Finally, last year a publication of Lazarifdis eta l. (DigDisSci 2017, 62:2348) showed the importance of the NLRP3 inflammasome complex in IBD patients already. NLRP3 was activated in 60% of the CD patients whereas no difference could be seen between UC and controls. These authors concluded that NLRP3 is activated in CD more than in UC. Concerning this present manuscript, the authors should discuss the controversial results in comparison to their observations.

Overall, the manuscript is quite interesting concerning the controversial results of IL-1beta and NLRP3. The staining methods should be completed with some new images in order to be more convincingly. Therefore, I propose a major revision.

Author Response

The authors analysed the expression of IL-1beta and NLRP3 in biopsies from IBD patients by qPCR analysis. They found upregulation of both in active UC and CD. Furthermore, immunostaining of colon paraffin sections were stained for NLRP3 and IL-1beta. Here, the quantification revealed a higher number of double+ cells in active UC as well as active CD. Less colocalisation is observed with NLRP3 and IL-1beta in active UC in comparison to normal and active CD patient sections.

First, the manuscript is more descriptive then explanatory. The authors should more try to undercover the mentioned blockade of caspase1 or proteinase-3 in relation with their patient biopsies.

Response 1: This study demonstrates the upregulation of NLRP3 and IL-1β in active IBD, describes the colonic localisation NLRP3 and IL-1β and highlights the reduced contribution of NLRP3 to IL-1β production in active UC. Overall this study is unique because it present an in vivo snapshot of protein activity in a background of changing lamina propria cellular dynamics. Examining the blockade of caspase-1 or proteinase-1 in the biopsies collected thus far is not possible and beyond the scope of this paper. It is envisaged that the results of this paper will direct future experiments examining the role of neutrophil-derived proteinases in active UC.

The house keeping gene EEF2 is used in Figure1. From literature it is known that this gene is stable in experimental models if colitis, but in human colon material it is rarely used. Why did this gene was used ?

Response 2: This is an important point raised by the reviewer and we acknowledge that that EEF2 is often overexpressed in gastrointestinal cancers [1], however none of the patients in this study were diagnosed with cancer at the time of biopsy collection. Furthermore, EEF2 has been used as a reference gene in human smooth muscle tissue [2] and more recently in human colonic biopsies [3].
In thisstudy we tested all biopsy samples for the 4 most common housekeeping genes
, , EEF2, GAPDH, HPRT1 and ACTB and compared the results across the 5 disease groups. All of the housekeeping genes demonstrated PCR amplification (Figure 1), however EEF2 showed slightly less variability (standard deviation = 0.2328) (Table 1) across the disease groups and similar PCR amplification to patient samples and therefore was the gene chosen for normalisation.

The quantification in Figure 2 is difficult to verify. Did the authors mean the blue shadowed cells or the brown cells ? The arrow shows somewhere in the section. What about the positive cells in the other sections ? Some more arrows should help to identify the positive cells. Same is concerning the Figure 3. Here no arrows were visible. Are some positive cells can be identified ?Helpful would be, if the authors can add the negative control stainings to the figures.

Response Point 2: In Figure 2 representative IHC images are presented to show the variability of NLRP3 expression in IBD. For clarification, blue is DAPI stain and show the cell nuclei, brown is DAB stain and indicates NLRP3 expression. The arrow presented in Figure 2 is merely there to direct the reader’s attention to a region where there is increased numbers of neutrophils within the lamina propria. Neutrophil influx is an important diagnostic feature of active UC and is consistent with the theme of this paper. In all biopsies, positive cells are indicated by the presence of brown staining. Negative control biopsies are not normally presented with IHC results in journal papers, however we have attached images of the NLRP3 and IL-1β antibody optimisation results for you to examine (see below).
NLRP3 and IL-1β Antibody Optimisation

In order to investigate the localisation of NLRP3 and IL-1β in colon tissue it was first necessary to establish the efficiency of the primary antibodies in human control tissues. The NLRP3 antibodies (ab17267 and ab16097, Abcam, Cambridge, MA, USA) demonstrated positive staining in the cytoplasmic regions of the stratified squamous epithelial layer of normal human tonsil (Figure 1). In agreement, distinct NLRP3 expression has previously been reported for the stratified non-keratinizing squamous epithelium of the oral and oesophageal mucosa [4].
The IL-1β antibody (ab9722, Abcam, Cambridge, MA, USA) demonstrated positive cytoplasmic staining for the circulating immune cells, such as macrophages, lymphocytes and dendritic cells,localised to the paracortical zone of human lymph node tissue (Figure 1). In
agreement, positive IL-1β staining has previously been reported within the germinal centre, paracortical and mantle zones of human lymph node tissue during infection [5].

In Figure 4, the optical density should be further explained. What unit do OD have ? How is the OD been calculated ?

Response 3: The reviewers comments are appreciated and in accordance the following text has been added to the original manuscript to describe the method for calculating the optical density in Figure 4 (lines 362-370). Since optical density is a ratio there are no units of measure.
Quantitative analysis of immunohistochemistry data was performed using the FIJI version (Dec 2009) of ImageJ software (http://imagej.nih.gov/ij/downloads.html, Rasband, W.S., ImageJ, U. S. National Institutes of Health, Bethesda, Maryland, USA), Microsoft Excel and GraphPad Prism (version 7, GraphPad Software Inc. CA. USA). Briefly, the mean intensity of the DAB signal was measured in FIJI, optical density was calculated in Excel using log (maximum intensity/mean intensity), where the maximum intensity of an 8-bit image = 255. Images containing partial tissue were excluded from analysis because of their potential to bias results by reducing the average optical density.

Concerning the Figure 5, the staining of NLRP3 is not optimal done. I would expect the signal not extracellular than intracellular. Here may be a positive control as well as a negative control would help to identify the right cells.

Response 4: The reviewers observations are appreciated. Good quality IF images are hard to acquire using paraffin embedded sections in part due to background tissue. Additionally some picture clarity is lost with the conversion of Word to PDF. IHC is considered a more powerful tool to examine protein expression because it allows for the identification of cell types and gives a clearer picture of the tissue architecture (similar to what a pathologist see and report on an H&E film). Protein expression in the colon is also easier to quantify with IHC DAB staining because tissue boarders are clearer. Cell membranes are harder to distinguish with IF however, IF is necessary for colocalisation studies. The data presented in Figure 5 merely indicates the overlap of fluorescence signal from NLRP3 and IL-1β and is not intended to identify cell types as this data has been presented in Figures 2 and 3.

Finally, last year a publication of Lazarifdis eta l. (DigDisSci 2017, 62:2348) showed the importance of the NLRP3 inflammasome complex in IBD patients already. NLRP3 was activated in 60% of the CD patients whereas no difference could be seen between UC and controls. These authors concluded that NLRP3 is activated in CD more than in UC. Concerning this present manuscript, the authors should discuss the controversial results in comparison to their observations.

Response 5: The paper: Activation of NLRP3 Inflammasome in Inflammatory Bowel Disease: Differences Between Crohn’s Disease and Ulcerative Colitis, Lazaros-Dimitrios Lazaeidis et al, 2017, DigDis Sci 62:2348-2356 describes a study using PBMC, with LPS stimulation in the presence or absence of MSU. Our study was performed using human colonic biopsies from active and quiescent disease and therefore the results cannot be compared and are not considered controversial.
References:

1. Nakamura, J.,Aoyagi, S.,Nanchi, I.,Nakatsuka, S.,Hirata, E.,Shibata, S.,Fukuda, M.,Yamamoto, Y.,Fukuda, I.,Tatsumi, N.,Ueda, T.,Fujiki, F.,Nomura, M.,Nishida, S.,Shirakata, T.,Hosen, N.,Tsuboi, A.,Oka, Y.,Nezu, R.,Mori, M.,Doki, Y.,Aozasa, K.,Sugiyama, H.. Oji, Y., Overexpression of eukaryotic elongation factor eEF2 in gastrointestinal cancers and its involvement in G2/M progression in the cell cycle. Int J Oncol 2009, 34, (5), 1181-9.
2. Myers, S. A.,Nield, A.,Chew, G. S.. Myers, M. A., The zinc transporter, Slc39a7 (Zip7) is implicated in glycaemic control in skeletal muscle cells. PloS one 2013, 8, (11), e79316.
3. Ranson, N.,Veldhuis, M.,Mitchell, B.,Fanning, S.,Cook, A. L.,Kunde, D.. Eri, R., Nod-Like Receptor Pyrin-Containing Protein 6 (NLRP6) Is Up-regulated in Ileal Crohn's Disease and Differentially Expressed in Goblet Cells. Cell Mol Gastroenterol Hepatol 2018, 6, (1), 110-112 e8.
4. Kummer, J. A.,Broekhuizen, R.,Everett, H.,Agostini, L.,Kuijk, L.,Martinon, F.,van Bruggen, R.. Tschopp, J., Inflammasome components NALP 1 and 3 show distinct but separate expression profiles in human tissues suggesting a site-specific role in the inflammatory response. The journal of histochemistry and cytochemistry : official journal of the Histochemistry Society 2007, 55, (5), 443-52.
5. Doitsh, G.,Galloway, N. L.,Geng, X.,Yang, Z.,Monroe, K. M.,Zepeda, O.,Hunt, P. W.,Hatano, H.,Sowinski, S.,Munoz-Arias, I.. Greene, W. C., Cell death by pyroptosis drives CD4 T-cell depletion in HIV-1 infection. Nature 2014, 505, (7484), 509-14.

Overall, the manuscript is quite interesting concerning the controversial results of IL-1beta and NLRP3. The staining methods should be completed with some new images in order to be more convincingly. Therefore, I propose a major revision.

Sincerely thank you for your valuable comments on our manuscript.

Round 2

Reviewer 2 Report

The authors answered to all critical points and delivered answers. I agree with the most points or the authors persuaded me to accept the argument. But one point can be still improved

The black arrow in figure 2 is still vexing me and i propose to replace the arrow to a maybe shaded area. The authors want to attract the attention to the increased number of neutrophils in the LP and therefore a dotted or broken line around the ROI would be more attractive instead of pointing somewhere in the middle.

Overall, the manuscript is now in better status and the authors addressed me points, except the vexing problem in figure 2. Therefore, I propose a minor revision.

Author Response

Response to Reviewer 2 Comments, Round 2:

Point 1: The  black arrow in Figure 2 is still vexing me and I propose to replace the arrow to a maybe shaded area. The authors want to attract the attention to the increased number of neutrophils in the LP and therefore a dotted  or broken line around the ROI would be more attractive instead of pointing somewhere in the middle.

Response 1: The reviewers comments are appreciated.  The arrow in Figure 2 has been removed and replaced with a broken line box which indicates more clearly the area of neutrophil influx. Likewise corresponding text on line 139 and 148 now reads: dotted black box instead of arrow.